# International comparison of China's digitalization level and its enlightenment

**Zongyuan Huang** *, **Miaomiao Qin**

School of Economics and Management, Beijing Jiaotong University, Beijing, China

* zyhuang@bjtu.edu.cn

**Data Availability Statement:** All relevant data are within the manuscript and its Supporting Information files.

**Funding:** The authors received no specific funding for this work.

## Abstract

The global digital wave has flourished in recent years, and the digital technology revolution has emerged. Digitalization plays an undeniable role in promoting modern economic and social development in multiple aspects such as economy, society, innovation, public services and sustainable development, China's digitalization application is also developing rapidly. In order to better measure the current situation of China's digitalization development level, this paper constructs a comprehensive evaluation index system of digitalization development from four dimensions of talents in the digital field, digital infrastructure construction, digitalization innovation ability and international competitiveness, and tests the index system. The entropy method is used to measure the level of digitalization development between China and the United States, the United Kingdom, France and other major developed countries in the world, and the coefficient of variation method, kernel density estimation and Dagum Gini coefficient method are used to analyze the temporal and spatial characteristics and regional differences of digitalization development level of seven countries. This paper makes a comparative analysis between China and major developed countries from the historical perspective of the evolution of the techno-economic paradigm. With a view to summarizing and exploring from it and drawing on the advanced experiences accumulated by the developed countries over a long period of time, so as to provide China with useful insights and bases for further enhancing its digitalization development level.

## 1. Introduction

With the development of information technology, the important role of digitization in modern economic development is becoming increasingly prominent. The so-called digitalization refers to a new economic paradigm that utilizes digital technology to promote the high-quality and sustainable development of the modern economy and society. For modern economic development, digitalization can greatly improve economic efficiency and resource utilization efficiency by improving production and operational processes. Through automation and intelligence, costs can be reduced, time and labor demand can be reduced, and more accurate predictions and plans can be achieved. Digitalization can also change the form of products and services, provide personalized solutions, and bring more innovation opportunities for enterprises and organizations, creating new business models and market opportunities, thereby

**Competing interests:** The authors have declared that no competing interests exist.

promoting innovation and adding value. Digitalization can also enhance value creation by increasing data-driven insights and decision-making capabilities through data analysis and mining. In addition, digitalization plays a better role in expanding markets and reducing transaction costs, promoting employment and human resource development, and supporting sustainable development, which means that it can promote economic development and form more efficient production and management methods, increase opportunities for innovation and growth, expand market potential, and support sustainable development, thereby bringing about a stronger and more competitive economic system.

The concept of a "digital economy" was first explicitly proposed at the G20 Hangzhou Summit in 2016 [1]. Since then, digitalization has rapidly become a highly valued theme in the economic development strategies of countries around the world. The digital strategic layout of major countries in the world is mainly reflected in: in terms of infrastructure construction, the United States has vigorously established advanced digital infrastructure, including high-speed broadband networks and full coverage of 5G technology, created a good digital environment by providing fast and reliable internet connectivity, and promoted research and development of cutting-edge digital technologies to consolidate the innovation advantages of key core technologies in the digital economy; in 2018, the German federal government put forward the construction of digital infrastructure, digital transformation innovation action, and full integration of digital technology into the innovation strategy; Japan continues to improve the digital talent training system, focusing on innovation in infrastructure and science and technology; the British government continues to activate the innovation effectiveness, focusing on data acquisition, talent training, research transformation and industry development [2]. It can be seen that there are commonalities in the digitalization strategy layout of major countries in the world, in which the three aspects of infrastructure construction, talent training and scientific and technological innovation play an important role. Taking this as the starting point, this paper compares and analyzes the digitalization development level between China and major developed countries by constructing a comprehensive evaluation index system for digitalization development, and the research aims to clarify the current situation of digitalization development in China.

Currently, countries around the world have launched digitalization development strategies to seize important opportunity windows. Developed countries such as Europe, the United States and Japan have accumulated long-term experience, while China is in the catch-up stage [3]. In view of this, this paper reviews and analyzes the evolution of the techno-economic paradigm in major developed countries and China, with a view to clearly understanding the gaps and shortcomings in China on the basis of learning from the advanced experience of other countries, so as to find out the effective ways and countermeasures of China's digitalization development.

## 2. Literature review

### 2.1 Relevant research on digitalization development

To accurately evaluate the level of digitalization development in various countries, domestic and foreign scholars have conducted discussions and analyses from different perspectives. For example, Andres et al. studied the development of the digital economy in 214 countries including China, Brazil, and France from 1990 to 2004 mainly from three aspects: factor endowment, population agglomeration, and Internet development [4]. Chinn and Fairlie measured the development level of the digital economy in 161 countries from 1990 to 2001 in terms of income, regulation, and infrastructure [5]. Wunnava and Leiter respectively studied the current status of the digital economy in various countries from three dimensions:

telecommunications infrastructure, sound implementation of economic laws and regulations, and personal quality [6]. Crenshaw and Robison studied the development level of the digital economy in various countries from three aspects: foreign investment, major city clusters, and manufacturing exports [7]. Lam et al. used interpolation simulation methods to study the development of Vietnam's digital economy from 1997 to 2002 from four aspects: Internet popularity, technical equipment, legal factors, and historical factors [8]. Domestic scholars have also conducted research on the evaluation of the digital economy from multiple perspectives. For example, Pan Wei and Han Botang conducted research from five aspects: total economic output, labor capital input, R&D capital input, urbanization level, and information technology factors, to analyze the digitalization level of urban agglomerations in the Beijing, Tianjin, Hebei, Yangtze River Delta and Pearl River Delta urban agglomerations [9]. Yang Daoling and Li Xiangli studied the digital economy development level of 64 countries along the "Belt and Road" except China in 2016 from three aspects of ICT foundation, ICT application, and ICT industry by using the dimensionless method and grading method [10]. Zhang Bochao and Shen Kaiyan analyzed the development level of the digital economy in 42 countries along the Belt and Road, including China, Mongolia and Singapore, from three perspectives: factor endowment and infrastructure, information technology, business and innovation environment [11]. Dong Youde and Mi Xiaoxiao measured the development level of the digital economy in 104 Asian and European countries from 2009 to 2016, including China and Russia, from three dimensions: system and innovation environment, infrastructure construction and information technology application [12]. In addition, domestic scholars also have conducted an evaluation and analysis of regional digital capabilities and industrial digital transformation [13–15], the digital transformation of enterprises is mostly based on theoretical analysis [16]. From this, it can be seen that there are still few studies on the comparative analysis of the digitalization level between China and the major developed countries, which is unfavorable for our country to better promote the digitalization development level and speed up the development of the digital economy. Starting from a comparative analysis of the digitalization development level between China and major developed countries around the world, this article delves into and analyzes the main problems that still exist in China's digitalization development. Starting from a new perspective of the synergy between technological economic paradigms and digital economic development, it seeks effective ways and methods to further enhance China's digitalization development. This will be beneficial for further promoting the high-quality development of China's digital economy.

## 2.2 Measurement of digitalization level

From the existing research results, domestic and foreign scholars have regarded digital technology as an important index for measuring the level of digitalization [17–21], although some scholars use a single index for evaluation [22, 23]. More scholars believe that the level of digitalization should be measured from multiple dimensions [24–33]. However, from the evaluation of national digitalization development level, there is currently no recognized measurement standard and unified evaluation system, and there are few targeted comparative analysis studies on the digitalization development level between China and major developed countries.

There are two difficulties in evaluating the digitalization development level in various countries: one is how to select reasonable and feasible indices, and the other is how to determine the weight of the evaluation index system through correct and objective weighting. On the basis of drawing on previous research results and considering factors such as the scientificity, effectiveness, rationality, reliability, and accessibility of indices and data, this paper measures the digitalization development level in China and the world's major developed countries from

the four dimensions of talents in the digital field, digital infrastructure construction, digitalization innovation capability and international competitiveness, which are characterized by highlighting the significance in the process of digitalization, and these indices have a certain portrayal ability for depicting the application level of digital technology.

## 3. Research methods

The direct use of digital technology to form indices is certainly the most ideal, but due to the lack of effective statistical data in this area, most scholars in practical research work adopt a comprehensive index system method composed of relevant data. Based on the existing research results, this paper constructs a comprehensive evaluation index system for digitalization development, which consists of 4 first-level indices and 12 second-level indices, including talents in the digital field, digital infrastructure construction, digitalization innovation ability and international competitiveness (see Table 1 for details). The data used in this paper are from the World Bank database, the United Nations Conference on Trade and Development, the World Trade Organization database, and the China National Statistical Yearbook, and the data period is 2012–2021. In order to make the table more beautiful, we use the letters A-D to arrange the primary and secondary indices. The letters in the following table are shown in Table 1.

In terms of talents in the digital field, the importance of talents in the digital field for digitalization development cannot be ignored. Therefore, this paper selects the two indices of R&D researchers per million people and education expenditure as a percentage of GDP to reflect a country's situation and level in terms of talents in the digital field from the perspectives of human resources and education funds respectively. In terms of digital infrastructure construction, digital devices play an important role in the construction of digital infrastructure to

**Table 1. Comprehensive evaluation index system for digitalization development.**

| Primary indices | Secondary indices | Index attribute | Index unit |
|---|---|---|---|
| Talents in the digital field (A) | R&D researchers per million people (A1) | Forward direction | Person |
| | Education expenditure as a percentage of GDP (A2) | Forward direction | % |
| Digital infrastructure construction (B) | Number of mobile phone users (B1) | Forward direction | Households per hundred people |
| | Number of fixed broadband subscribers (B2) | Forward direction | Households per hundred people |
| | Number of mobile broadband subscribers (B3) | Forward direction | Households per hundred people |
| | Internet penetration rate (B4) | Forward direction | % |
| Digitalization innovation capability (C) | Number of patent applications accepted (C1) | Forward direction | Items per thousand people |
| | Number of patent applications granted (C2) | Forward direction | Items per thousand people |
| | Telecommunications, computer and information services percentage change (year-on-year) (C3) | Forward direction | % |
| | ICT frontier technology readiness index (C4) | Forward direction | —— |
| International competitiveness (D) | Information and communication technology (ICT) product exports (percentage of total product exports)(D1) | Forward direction | % |
| | Digitally-deliverable services exports percentage of total trade in services (D2) | Forward direction | % |

connect users and infrastructure, support data transmission and processing, and promote cloud computing services and innovative digital products and services. Therefore, this paper selects the number of mobile phone users, the number of fixed broadband subscribers, the number of mobile broadband subscribers and the Internet penetration rate as four secondary indices to measure digital infrastructure construction. In terms of digitalization innovation capacity, digitalization innovation capacity is the key to achieve digitalization sustainable development. Therefore, this paper selects the number of patent applications accepted and the number of patent applications granted to evaluate a country's potential for scientific and technological innovation, and telecommunications, computer and information services percentage change, and ICT frontier technology readiness index to evaluate the conditions and potential of a country in the construction of digital economy and information society. In terms of international competitiveness, information and communication technology (ICT) product exports percentage of total product exports and digitally-deliverable services exports percentage of total trade in services can measure digitization level of goods and services from an export perspective.

In order to evaluate the quality, stability and internal consistency of the constructed index system, this paper carries out the Kendall's W test on 12 indices of digitization development, which helps to determine the reliability and validity of the index system, and makes necessary adjustments and improvements according to the test results for use in subsequent analysis and decision-making. The calculation formula is as follows:

$$W = \frac{12S}{m^2(N^3 - N)} \quad (1)$$

$$S = \sum\nolimits_{j=1}^{m}(R_i - \bar{R})^2, \bar{R} = \frac{1}{N}\sum\nolimits_{i=1}^{N}R_i, R_i = \sum\nolimits_{j=1}^{m}R_{ij} \quad (2)$$

Where, $m$ is the number of evaluators, N is the sample size, and $R_i$ is the rank sum.

The results of Kendall's W coefficient and P-value for 12 indices in seven countries were calculated as shown in Table 2.

Kendall coefficient consistency test results show that the significance P-value of the overall data of China, the United States, the United Kingdom, France, Germany, Japan and Korea are all 0.000***, which presents significance at the level and rejects the original hypothesis, so the data present consistency. Meanwhile, the Kendall coordination coefficient W values of the model are 0.986, 0.985, 0.963, 0.964, 0.982, 0.987 and 0.991, which are above 0.8, indicating excellent consistency. This therefore indicates that the selected indices have a high degree of consistency among the seven countries and can be used to compare and evaluate the performance of the seven countries on these indices.

**Table 2. Kendall's W test results for 12 indices in seven countries.**

| Country | Index name | Kendall's W coefficient | P |
|---------|------------|-------------------------|---|
| CHN | A1, A2, B1, B2, B3, B4, C1, C2, C3, C4, D1, D2 | 0.986 | 0.000*** |
| USA | A1, A2, B1, B2, B3, B4, C1, C2, C3, C4, D1, D2 | 0.985 | 0.000*** |
| UK | A1, A2, B1, B2, B3, B4, C1, C2, C3, C4, D1, D2 | 0.963 | 0.000*** |
| FRA | A1, A2, B1, B2, B3, B4, C1, C2, C3, C4, D1, D2 | 0.964 | 0.000*** |
| GER | A1, A2, B1, B2, B3, B4, C1, C2, C3, C4, D1, D2 | 0.982 | 0.000*** |
| JPN | A1, A2, B1, B2, B3, B4, C1, C2, C3, C4, D1, D2 | 0.987 | 0.000*** |
| KR | A1, A2, B1, B2, B3, B4, C1, C2, C3, C4, D1, D2 | 0.991 | 0.000*** |

## 1. Entropy value method

As a commonly used decision analysis method, the entropy method has a strong comprehensive analysis ability for complex decision problems, especially for multi-attribute decision problems such as digital development level evaluation, and can better avoid the bias caused by subjective influence. It considers multiple evaluation indices comprehensively and normalizes the entropy value obtained from each selection scheme, thus making the results more comparable and intuitive, which is convenient for decision-makers to make decisions.

### (1) Standardized processing

The different units of each index in the evaluation index system of the digitalization development level will affect the research results. Firstly, it is necessary to standardize the original data before calculating. If the extreme value "0" appears after standardization, 0.001 translation processing must be carried out on the standardized value in order to eliminate the impact of the extreme value "0" in the subsequent calculation process, namely $Y_{ij} = Y'_{ij} + d$, $d = 0.001$.

Standardization of positive indices:

$$Y_{ij} = \frac{X_{ij} - minX_{ij}}{maxX_{ij} - minX_{ij}} \tag{3}$$

Among them, $X_{ij}$ and $Y_{ij}$ respectively represent the initial data and the standardized data. $Y_{ij} \in [0,1]$, $maxX_{ij}$ and $minX_{ij}$ represent the maximum and minimum values in the index. Where $i = 1, \ldots 2 \ldots m$ represents the year, $j = 1, 2 \ldots n$ indicates the number of indices.

Standardization of negative indices:

$$Y_{ij} = \frac{maxX_{ij} - X_{ij}}{maxX_{ij} - minX_{ij}} \tag{4}$$

### (2) Entropy method for determining index weights

In the first step, calculate the proportion of the ith year under the jth index $P_{ij}$:

$$P_{ij} = \frac{Y_{ij}}{\sum\limits_{i=1}^{m} Y_{ij}} \tag{5}$$

In the second step, the entropy value of j indices is calculated: $E_j$

$$E_j = -k \sum\limits_{i=1}^{m} p_{ij} \ln p_{ij} \tag{6}$$

$k > 0$, ln is the natural logarithm, and the constant k is related to the number of years, such that $k = \frac{1}{\ln m}$, $E_j \in [0,1]$.

Step 3: Calculate the difference of the jth index $D_j$:

$$D_j = 1 - E_j \tag{7}$$

Step 4: Calculate the weight of the jth index: $W_j$

$$W_j = \frac{D_j}{\sum_{j=1}^{n} D_j} \tag{8}$$

(3) Calculate the comprehensive index of digitalization development level

The weighted summation method is used to calculate the comprehensive evaluation index $Y_i$:

$$Y_i = \sum_{j=1}^{n} W_j X_{ij} \tag{9}$$

## 2. Coefficient of variation method

The coefficient of variation is a measure of the degree of dispersion of an observed series, expressed as the standard deviation divided by the mean of the data set. The coefficient of variation method is often used to compare the differences between two or more data sets, which can more objectively evaluate the variability of different data sets for effective comparison and analysis. Therefore, the coefficient of variation is chosen as another index to measure the differences in digitalization development in this paper. The higher the value of variation coefficient, the greater the degree of data dispersion, indicating that a country's digitalization development level is more unbalanced. The coefficient of variation can be divided into five levels according to the degree of fluctuation, as shown in Table 3.

## 3. Kernel density estimation

Kernel density estimation is a non-parametric estimation method that can be used to analyze changes in digitalization development over time by fitting the trend of a variable to its probability density function. Suppose its probability density function is $f$, and the kernel density is estimated as:

$$f_h(x) = \frac{1}{n} \sum_{i=1}^{n} K_h(x - x_i) = \frac{1}{nh} \sum_{i=1}^{n} K(\frac{x - x_i}{h}) = \frac{1}{h} \times \sum_{i=1}^{n} \left( \frac{1}{\sqrt{2\pi}\sigma h} e^{\frac{-(x-x_i)^2}{2\sigma^2 h^2}} \right) \tag{10}$$

K (x) is the kernel function, h>0 is a smoothing parameter, called bandwidth; and $\sigma$ is the sample standard deviation.

## 4. Dagum Gini coefficient and its decomposition method

**Table 3. Classification of variation coefficient and fluctuation level.**

| Variation coefficient | Fluctuation level |
|---|---|
| CV<0.05 | low fluctuation |
| 0.05≤CV<0.1 | lower fluctuation |
| 0.1≤CV<0.15 | moderate fluctuation |
| 0.15≤CV<0.2 | higher fluctuation |
| CV≥0.2 | high fluctuation |

Dagum (1997) proposed the Gini coefficient decomposition method by subgroup, which effectively solved the problem of the source of spatial differentiation and cross-overlap among subsamples, and could examine the degree of spatial differentiation of digitalization development in seven countries. The Dagum Gini coefficient method is used in this paper to analyze the differences in digitalization development of seven countries. The basic definition of Gini coefficient is shown in Eq (11), where $y_{ji}$ ($y_{hr}$) is the digitalization development of a country in the region of $j(h)$, represents the mean value of the digitalization development of each country, n represents the number of countries, k is the number of regions, and $nj(nh)$ is the number of countries in the $j(h)$ region.

$$G = \sum_{j=1}^{k} \sum_{h=1}^{k} \sum_{i=1}^{nj} \sum_{r=1}^{nh} \frac{|y_{ji} - y_{hr}|}{2n^2\bar{y}} \tag{11}$$

In the process of Gini coefficient decomposition, the mean value of digitalization development in each region needs to be calculated separately and sorted, namely:

$$\bar{Y}_1 \leq \cdots \leq \bar{Y}_h \leq \cdots \leq \bar{Y}_k \tag{12}$$

The Gini coefficient G representing the aggregate can thus be further decomposed into the sum of the intra-regional (within-group) variance contribution Gw, the inter-regional (between-group) variance contribution Gb, and the super variable density contribution Gt. The overall differences are further decomposed and represented by the following equations.

$$G_{jj} = \frac{\frac{1}{2\bar{Y}_j} \sum_{i=1}^{nj} \sum_{r=1}^{nj} |y_{ji} - y_{hr}|}{n_j^2} \tag{13}$$

$$G_w = \sum_{j=1}^{k} G_{jj} p_j s_j \tag{14}$$

$$G_{jh} = \frac{\sum_{i=1}^{nj} \sum_{r=1}^{nj} |y_{ji} - y_{hr}|}{n_j n_h (\bar{Y}_j + \bar{Y}_h)} \tag{15}$$

$$G_{nb} = \sum_{j=2}^{k} \sum_{h=1}^{j-1} G_{jh} (p_j s_h + p_h s_j) D_{jh} \tag{16}$$

$$G_t = \sum_{j=2}^{k} \sum_{h=1}^{j-1} G_{jh} (p_j s_h + p_h s_j)(1 - D_{jh}) \tag{17}$$

Eq (13) reflects the Gini coefficient Gjj of digital development in region j. Eq (14) represents the contribution rate Gw of intra-regional differences in the region to the overall differences. Eq (15) reflects the inter-group Gini coefficient Gjh between region j and region h. Eq (16) reflects the contribution rate Gnb of the inter-regional net worth differences to the overall difference. Formula (17) describes the contribution $G_t$ of super-variable density. In general, the smaller the intra-group Gini coefficient, the closer the gap between countries within the group. The larger the intra-group Gini coefficient, the greater the gap between countries within the group.

## 4. Data results and analysis

### 4.1 Data results and analysis

The weights of 12 indices in the four dimensions of talents in the digital field, digital infrastructure construction, digitalization innovation capability and international competitiveness are calculated respectively by the Formulas in steps (1) and (2) of the entropy method (see Table 4). Finally, the composite index scores of four dimensions and the total composite index scores of digitalization development as well as the trend of changes were obtained (see Tables 5 and 6).

In order to analyze the problem more clearly and intuitively, the scatter plot is drawn according to the total score of comprehensive digitalization development index of the seven countries in Table 6, as shown in Fig 1.

The total score of digitalization development is calculated from the comprehensive index scores of the four dimensions of talents in the digital field, digital infrastructure construction, digitalization innovation capability and international competitiveness of the seven countries in Table 5. It can be seen from Table 6 that the United States ranked first in digitalization development level among the seven countries in 2020 and 2021, and it can be seen from Fig 1 that the country's balance of digitalization development level from 2012 to 2021 is better. The UK had a higher level of digitalization development in 2012, but the overall fluctuation was large, which can be seen more intuitively from Fig 1. France showed a year-on-year increase in the digitalization development level from 2012 to 2020, and the total score of digitalization development in Germany ranked third among the seven countries. However, it can be seen from Fig 1 that Germany's digitalization development in the decade from 2012 to 2021 is extremely uneven. Japan's digitalization development level is lower than that of the United States, the United Kingdom, France and Germany, but higher than that of Korea and China, which can be seen in Fig 1 as an uneven development. Except for 2019, Korea's digitalization development level showed an increasing trend year by year from 2014 to 2021, but the country's digitalization development is poorly balanced as shown in Fig 1. China's digitalization development level is the lowest among the seven countries, and its development is relatively slow. Fig 1 showed that China's digitalization development is also poorly balanced.

As can be seen from Table 5, in terms of talents in the digital field, the United States and France scored the highest among the seven countries. From 2012 to 2014, the United States also scored the highest among the seven countries in terms of talents in the digital field, which is related to the fact that the United States has taken the development of talents in the digital field as an important national strategy earlier and has made outstanding achievements in this field. Compared to other countries, China and Korea scored lower in terms of talents in the digital field. In terms of digital infrastructure construction, China, the United States, Japan and Korea has shown a year-on-year increase in the level of digital infrastructure development from 2012 to 2021, but China's digital infrastructure construction level is the lowest among the seven countries, indicating that there is still a certain gap between China and other countries in this respect. In terms of digitalization innovation capability, China has the lowest score, indicating that there is still a large gap between China and the other six countries in this respect, which can be more intuitively reflected in Fig 2. From Fig 2, it can be seen that China's digitalization innovation capability has significantly improved in 2021 compared to 2020. On the whole, from 2012 to 2021, Japan, Korea, Germany and the United Kingdom experienced significant fluctuations in their digitalization innovation capabilities, while the United States and France have a substantial increase in their digitalization innovation capabilities.

As can be seen from Table 5, China also had the lowest score in terms of international competitiveness, and the United States ranked first. In 2020 and 2021, the United States has already reached a higher level than that of other six countries in international competitiveness.

**Table 4. Calculation results of digitalization development evaluation indices in seven countries: China, the United States, the United Kingdom, France, Germany, Japan and Korea.**

| Index categories | Index name | Entropy value | | | | | | | Coefficient of variation | | | | | | | Weight | | | | | | |
|---|---|---|---|---|---|---|---|---|---|---|---|---|---|---|---|---|---|---|---|---|---|---|
| | | CHN | USA | UK | FRA | GER | JPN | KR | CHN | USA | UK | FRA | GER | JPN | KR | CHN | USA | UK | FRA | GER | JPN | KR |
| A | A1 | 0.827 | 0.879 | 0.916 | 0.878 | 0.847 | 0.922 | 0.862 | 0.173 | 0.121 | 0.084 | 0.122 | 0.153 | 0.078 | 0.138 | 0.077 | 0.069 | 0.064 | 0.099 | 0.102 | 0.046 | 0.072 |
| | A2 | 0.78 | 0.824 | 0.904 | 0.934 | 0.937 | 0.842 | 0.944 | 0.22 | 0.176 | 0.096 | 0.066 | 0.063 | 0.158 | 0.056 | 0.097 | 0.101 | 0.073 | 0.053 | 0.042 | 0.093 | 0.029 |
| B | B1 | 0.891 | 0.908 | 0.891 | 0.892 | 0.922 | 0.891 | 0.898 | 0.109 | 0.092 | 0.109 | 0.108 | 0.078 | 0.109 | 0.102 | 0.048 | 0.053 | 0.083 | 0.088 | 0.053 | 0.065 | 0.054 |
| | B2 | 0.851 | 0.877 | 0.925 | 0.9 | 0.877 | 0.878 | 0.89 | 0.149 | 0.123 | 0.075 | 0.1 | 0.123 | 0.122 | 0.11 | 0.066 | 0.070 | 0.057 | 0.081 | 0.082 | 0.072 | 0.058 |
| | B3 | 0.891 | 0.888 | 0.887 | 0.904 | 0.907 | 0.821 | 0.868 | 0.109 | 0.112 | 0.113 | 0.096 | 0.093 | 0.179 | 0.132 | 0.048 | 0.064 | 0.086 | 0.078 | 0.062 | 0.106 | 0.069 |
| | B4 | 0.874 | 0.864 | 0.915 | 0.909 | 0.905 | 0.933 | 0.889 | 0.126 | 0.136 | 0.085 | 0.091 | 0.095 | 0.067 | 0.111 | 0.056 | 0.078 | 0.065 | 0.073 | 0.063 | 0.039 | 0.058 |
| C | C1 | 0.857 | 0.904 | 0.904 | 0.896 | 0.914 | 0.871 | 0.917 | 0.143 | 0.096 | 0.096 | 0.104 | 0.086 | 0.129 | 0.083 | 0.064 | 0.055 | 0.073 | 0.085 | 0.058 | 0.077 | 0.044 |
| | C2 | 0.745 | 0.878 | 0.865 | 0.879 | 0.872 | 0.921 | 0.881 | 0.255 | 0.122 | 0.135 | 0.121 | 0.128 | 0.079 | 0.119 | 0.113 | 0.070 | 0.103 | 0.098 | 0.086 | 0.047 | 0.062 |
| | C3 | 0.773 | 0.899 | 0.897 | 0.907 | 0.944 | 0.871 | 0.844 | 0.227 | 0.101 | 0.103 | 0.093 | 0.056 | 0.129 | 0.156 | 0.101 | 0.058 | 0.078 | 0.075 | 0.038 | 0.076 | 0.082 |
| | C4 | 0.418 | 0.819 | 0.78 | 0.829 | 0.702 | 0.607 | 0.315 | 0.582 | 0.181 | 0.22 | 0.171 | 0.298 | 0.393 | 0.685 | 0.258 | 0.103 | 0.167 | 0.139 | 0.200 | 0.232 | 0.359 |
| D | D1 | 0.925 | 0.866 | 0.909 | 0.929 | 0.906 | 0.889 | 0.895 | 0.075 | 0.134 | 0.091 | 0.071 | 0.094 | 0.111 | 0.105 | 0.033 | 0.077 | 0.069 | 0.058 | 0.063 | 0.066 | 0.055 |
| | D2 | 0.915 | 0.649 | 0.89 | 0.908 | 0.776 | 0.863 | 0.888 | 0.085 | 0.351 | 0.11 | 0.092 | 0.224 | 0.137 | 0.112 | 0.038 | 0.201 | 0.083 | 0.074 | 0.151 | 0.081 | 0.058 |

Table 5. Composite index scores of seven countries in each dimension from 2012 to 2021.

| Year | Talent scores in the digital field | | | | | | | Scores of digital infrastructure construction | | | | | | | Digitalization Innovation Capability Scores | | | | | | | International competitiveness scores | | | | | | |
|---|---|---|---|---|---|---|---|---|---|---|---|---|---|---|---|---|---|---|---|---|---|---|---|---|---|---|---|---|
| | CHN | USA | UK | FRA | GER | JPN | Kr | CHN | USA | UK | FRA | GER | JPN | Kr | CHN | USA | UK | FRA | GER | JPN | Kr | CHN | USA | UK | FRA | GER | JPN | Kr |
| 2012 | 0.098 | 0.101 | 0.073 | 0.039 | 0.031 | 0.093 | 0.001 | 0.0002 | 0.013 | 0.083 | 0.031 | 0.001 | 0.001 | 0.001 | 0.107 | 0.058 | 0.203 | 0.112 | 0.228 | 0.081 | 0.385 | 0.027 | 0.029 | 0.047 | 0.058 | 0.013 | 0.066 | 0.001 |
| 2013 | 0.060 | 0.107 | 0.062 | 0.063 | 0.030 | 0.101 | 0.032 | 0.021 | 0.020 | 0.134 | 0.063 | 0.050 | 0.043 | 0.019 | 0.012 | 0.064 | 0.220 | 0.123 | 0.245 | 0.283 | 0.445 | 0.040 | 0.011 | 0.021 | 0.065 | 0.001 | 0.045 | 0.014 |
| 2014 | 0.040 | 0.110 | 0.081 | 0.072 | 0.024 | 0.109 | 0.030 | 0.042 | 0.047 | 0.135 | 0.122 | 0.095 | 0.068 | 0.055 | 0.030 | 0.056 | 0.084 | 0.094 | 0.058 | 0.060 | 0.089 | 0.025 | 0.028 | 0.059 | 0.058 | 0.018 | 0.041 | 0.024 |
| 2015 | 0.061 | 0.034 | 0.082 | 0.070 | 0.059 | 0.047 | 0.043 | 0.069 | 0.083 | 0.148 | 0.105 | 0.116 | 0.089 | 0.086 | 0.066 | 0.117 | 0.143 | 0.113 | 0.111 | 0.065 | 0.072 | 0.031 | 0.062 | 0.063 | 0.071 | 0.051 | 0.050 | 0.041 |
| 2016 | 0.122 | 0.022 | 0.066 | 0.065 | 0.068 | 0.031 | 0.042 | 0.096 | 0.145 | 0.146 | 0.144 | 0.126 | 0.113 | 0.118 | 0.300 | 0.183 | 0.120 | 0.190 | 0.113 | 0.068 | 0.063 | 0.035 | 0.090 | 0.106 | 0.091 | 0.084 | 0.045 | 0.053 |
| 2017 | 0.040 | 0.056 | 0.073 | 0.093 | 0.091 | 0.038 | 0.054 | 0.127 | 0.161 | 0.124 | 0.185 | 0.163 | 0.167 | 0.153 | 0.052 | 0.228 | 0.313 | 0.211 | 0.335 | 0.315 | 0.097 | 0.045 | 0.090 | 0.094 | 0.056 | 0.096 | 0.045 | 0.075 |
| 2018 | 0.023 | 0.071 | 0.052 | 0.094 | 0.115 | 0.033 | 0.072 | 0.158 | 0.186 | 0.140 | 0.225 | 0.191 | 0.185 | 0.159 | 0.182 | 0.205 | 0.377 | 0.247 | 0.332 | 0.299 | 0.150 | 0.058 | 0.035 | 0.072 | 0.087 | 0.083 | 0.033 | 0.082 |
| 2019 | 0.036 | 0.082 | 0.070 | 0.086 | 0.145 | 0.051 | 0.092 | 0.184 | 0.213 | 0.200 | 0.267 | 0.214 | 0.221 | 0.177 | 0.189 | 0.247 | 0.282 | 0.273 | 0.330 | 0.409 | 0.107 | 0.046 | 0.037 | 0.092 | 0.059 | 0.082 | 0.033 | 0.079 |
| 2020 | 0.050 | 0.152 | 0.111 | 0.152 | 0.102 | 0.102 | 0.098 | 0.198 | 0.229 | 0.181 | 0.298 | 0.244 | 0.243 | 0.217 | 0.154 | 0.174 | 0.368 | 0.237 | 0.128 | 0.139 | 0.092 | 0.065 | 0.261 | 0.117 | 0.093 | 0.213 | 0.129 | 0.112 |
| 2021 | 0.044 | 0.117 | 0.091 | 0.121 | 0.124 | 0.076 | 0.095 | 0.217 | 0.265 | 0.218 | 0.317 | 0.244 | 0.252 | 0.239 | 0.219 | 0.245 | 0.244 | 0.262 | 0.174 | 0.101 | 0.177 | 0.027 | 0.228 | 0.084 | 0.030 | 0.183 | 0.116 | 0.104 |
| Mean value | 0.057 | 0.085 | 0.076 | 0.085 | 0.079 | 0.068 | 0.056 | 0.111 | 0.136 | 0.151 | 0.176 | 0.144 | 0.138 | 0.122 | 0.131 | 0.158 | 0.236 | 0.186 | 0.205 | 0.182 | 0.168 | 0.040 | 0.087 | 0.075 | 0.067 | 0.082 | 0.060 | 0.059 |

Table 6. Total scores of the comprehensive digitalization development index for seven countries from 2012 to 2021.

| Year | CHN | USA | UK | FRA | GER | JPN | Kr |
|------|------|------|--------|------|--------|------|------|
| 2012 | 0.232 | 0.201 | 0.406 | 0.241 | 0.272 | 0.240 | 0.386 |
| 2013 | 0.133 | 0.203 | 0.437 | 0.313 | 0.3252 | 0.472 | 0.510 |
| 2014 | 0.137 | 0.240 | 0.3597 | 0.346 | 0.196 | 0.278 | 0.198 |
| 2015 | 0.227 | 0.295 | 0.436 | 0.358 | 0.337 | 0.250 | 0.242 |
| 2016 | 0.552 | 0.441 | 0.438 | 0.490 | 0.391 | 0.257 | 0.275 |
| 2017 | 0.263 | 0.535 | 0.604 | 0.545 | 0.685 | 0.565 | 0.379 |
| 2018 | 0.420 | 0.496 | 0.641 | 0.653 | 0.720 | 0.550 | 0.463 |
| 2019 | 0.455 | 0.580 | 0.643 | 0.684 | 0.771 | 0.713 | 0.455 |
| 2020 | 0.467 | 0.815 | 0.778 | 0.780 | 0.687 | 0.613 | 0.518 |
| 2021 | 0.506 | 0.855 | 0.636 | 0.730 | 0.724 | 0.545 | 0.615 |
| Mean value | 0.339 | 0.466 | 0.538 | 0.514 | 0.511 | 0.448 | 0.404 |

In order to further compare the differences among the digitalization development of seven countries, this paper calculates the coefficient of variation value of 12 indices in seven countries, as shown in Table 7.

As can be seen from Table 7, from an overall perspective, China's coefficient of variation is relatively large, indicating that China's digitalization development level is more unbalanced, and that there are obvious imbalances in terms of talents in the digital field, digital infrastructure construction, digitalization innovation capability and international competitiveness. Compared with China, on the whole, other countries have smaller coefficient of variation values, indicating that these six countries have made more balanced and stable progress in digitalization development.

In addition, by observing the smoothness of the Gaussian kernel density estimation curve of a country's digitalization development, we can initially understand the equilibrium and stability of the country's digitalization level. The bandwidth in Gaussian kernel density estimation determines the smoothness of the estimation, that is, the wider the bandwidth, the smoother the curve of the kernel density estimation will be. In this paper, the kernel density estimation method is used to calculate the broadband h values of seven countries, and the kernel density estimation graph is plotted, as shown in Fig 3. Among the seven countries, the broadband h value of the United States is the largest, which is 0.159. It can be seen from Fig 3 that the digitalization development of this country presents a smooth kernel density distribution, indicating that the overall development level of this country in the digitalization field is relatively

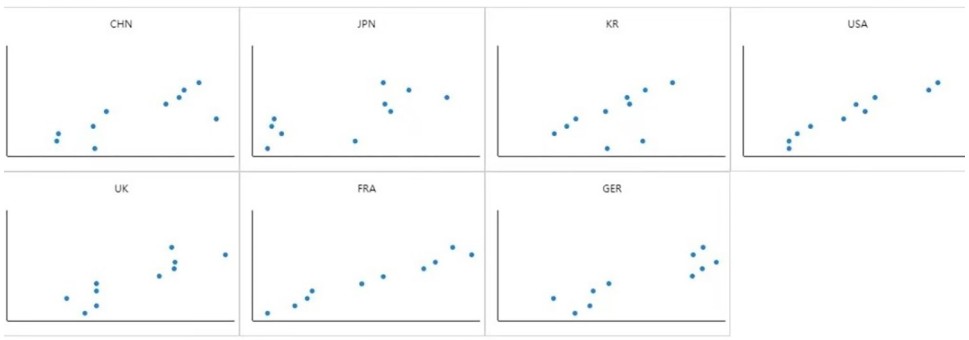

Fig 1. Scatter chart of digitalization development in seven countries.

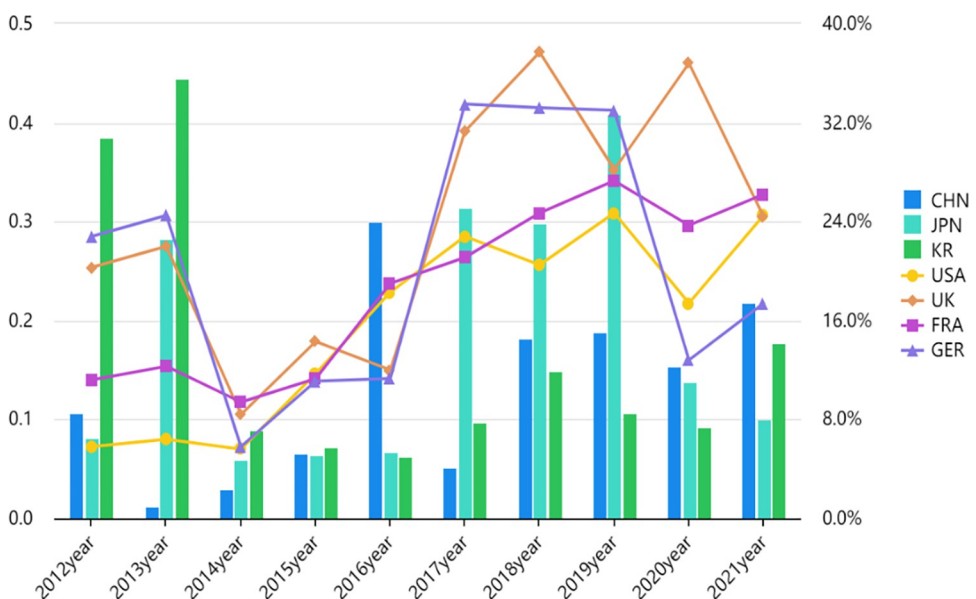

**Fig 2. Score chart of digitalization innovation capabilities in seven countries from 2012 to 2021.**

balanced, and the gap between various indices will not be too large, and the development is relatively stable. Germany, France, Japan, China, the United Kingdom and Korea have broadband h values of 0.150, 0.129, 0.118, 0.105, 0.093 and 0.089, respectively. If the broadband value is smaller, the kernel density distribution fluctuates greatly, which may mean that there is a large difference in the level of digitalization development, the development is not balanced or the development is in a violent fluctuation.

In order to measure the intra-group and inter-group gaps in the digitalization development of seven countries, Dagum Gini coefficient method is adopted in this paper and these gaps are analyzed in depth. China, Japan and South Korea are divided into Asia, the United Kingdom and the United States into both sides of the Atlantic, and Germany and France into Europe. Table 8 describes the overall, intra-regional and inter-regional Gini coefficients of the digitalization development index of seven countries. It can be found that the overall Gini coefficients

**Table 7. CV coefficient results for 12 indices in seven countries.**

| Index categories | Index name | CV coefficient | | | | | | |
|---|---|---|---|---|---|---|---|---|
| | | **CHN** | **USA** | **UK** | **FRA** | **GER** | **JPN** | **KR** |
| A | A1 | 0.227 | 0.073 | 0.055 | 0.069 | 0.091 | 0.025 | 0.119 |
| | A2 | 0.047 | 0.112 | 0.029 | 0.009 | 0.024 | 0.064 | 0.065 |
| B | B1 | 0.147 | 0.038 | 0.015 | 0.045 | 0.055 | 0.123 | 0.363 |
| | B2 | 0.366 | 0.082 | 0.06 | 0.077 | 0.098 | 0.08 | 0.065 |
| | B3 | 0.473 | 0.203 | 0.133 | 0.214 | 0.255 | 0.275 | 0.047 |
| | B4 | 0.185 | 0.099 | 0.03 | 0.03 | 0.028 | 0.049 | 0.055 |
| C | C1 | 0.339 | 0.099 | 0.55 | 0.059 | 0.088 | 0.071 | 0.192 |
| | C2 | 0.518 | 0.152 | 0.151 | 0.119 | 0.185 | 0.107 | 0.277 |
| | C3 | 0.991 | 0.606 | 1.191 | 1.96 | 1.524 | 0.894 | 0.602 |
| | C4 | 0.216 | 0.256 | 0.06 | 0.081 | 0.062 | 0.061 | 0.046 |
| D | D1 | 0.023 | 0.037 | 0.157 | 0.034 | 0.057 | 0.041 | 0.181 |
| | D2 | 0.121 | 0.125 | 0.059 | 0.042 | 0.054 | 0.152 | 0.235 |

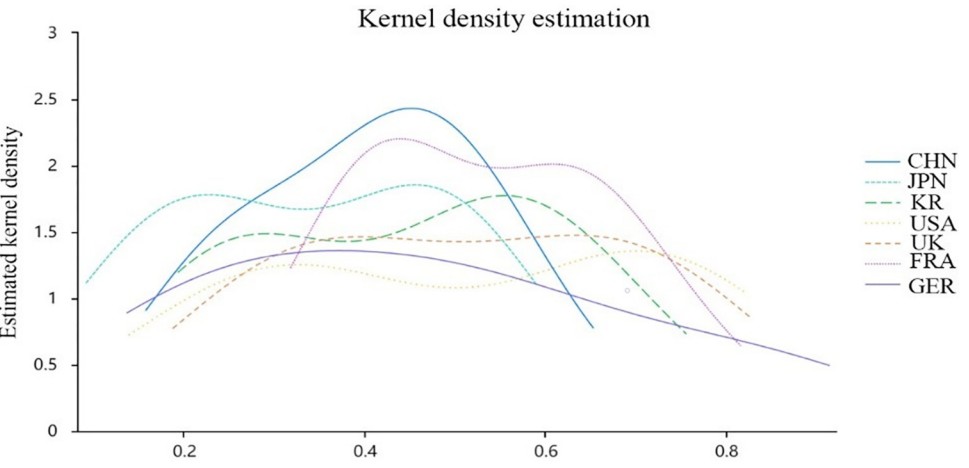

**Fig 3. Kernel density estimation graph for seven countries.**

of the digitalization development index of seven countries show a fluctuating and decreasing trend, experiencing an upward, a downward, a small upward, a downward, a small upward, and a downward trend during the period of 2012–2021. This indicates that the overall differences in the digitalization development of the seven countries are showing a narrowing trend, which means that the countries are gradually increasing the importance of digitalization development, and there is a significant effect of digitalization development in the seven countries by 2021.

As an important index for identifying cross-overlap phenomenon between regions, it can be seen from Fig 4 that the contribution rate of super variable density to the overall digitalization development in 2012, 2013 and 2016 was 46.91%, 54.55% and 39.93% respectively, indicating that intra-group and inter-group differences contributed significantly to the overall differences. In other years, the inter-group contribution is much greater than the intra-group contribution, which means that the regional differences in digitalization development in the seven countries mainly stem from inter-regional differences. The implication from the results of Gini coefficient and contribution rate is that the focus of regional differences in

**Table 8. Gini coefficient and contribution rate results of overall digitalization development.**

| Year | Gini coefficient | | | | Contribution rate | | |
|---|---|---|---|---|---|---|---|
| | Overall | Intra-group Gini coefficient Gw | Inter-group Gini coefficient Gb | Super variable density Gini coefficient Gt | Intra-group contribution rate Gw | Inter-group contribution rate Gb | Super variable density contribution rate Gt |
| 2012 | 0.138 | 0.039 | 0.034 | 0.065 | 28.48% | 24.61% | 46.91% |
| 2013 | 0.214 | 0.06 | 0.038 | 0.117 | 27.89% | 17.57% | 54.55% |
| 2014 | 0.17 | 0.045 | 0.088 | 0.037 | 26.34% | 51.86% | 21.80% |
| 2015 | 0.126 | 0.014 | 0.098 | 0.014 | 10.99% | 77.91% | 11.10% |
| 2016 | 0.137 | 0.035 | 0.048 | 0.055 | 25.35% | 34.73% | 39.93% |
| 2017 | 0.140 | 0.032 | 0.098 | 0.009 | 23.28% | 70.48% | 6.24% |
| 2018 | 0.103 | 0.017 | 0.082 | 0.004 | 16.59% | 79.62% | 3.80% |
| 2019 | 0.104 | 0.022 | 0.067 | 0.015 | 21.24% | 63.97% | 14.80% |
| 2020 | 0.106 | 0.013 | 0.093 | 0.000 | 12.18% | 87.71% | 0.12% |
| 2021 | 0.095 | 0.014 | 0.07 | 0.011 | 14.52% | 73.56% | 11.93% |

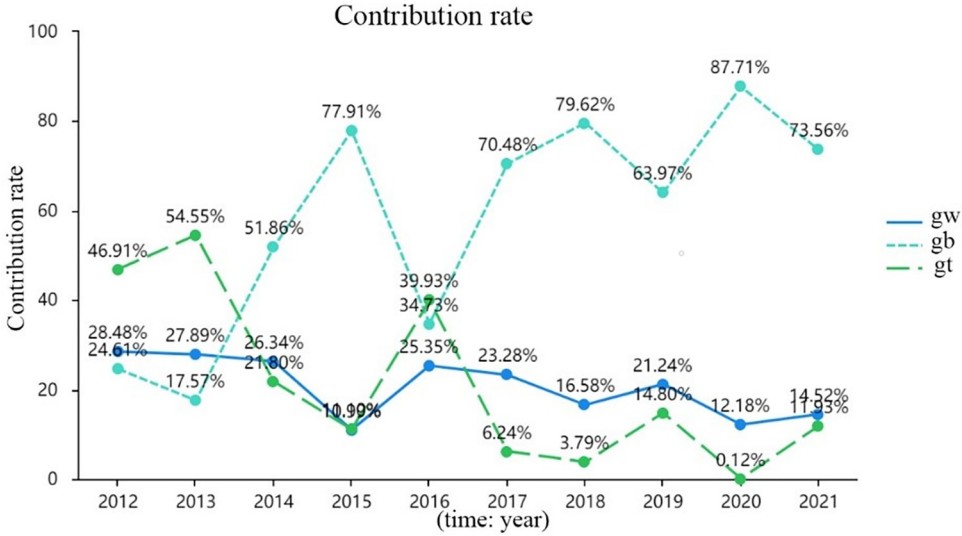

**Fig 4. Overall contribution rate of digitalization development.**

digitalization in seven countries should be the coordinated development of digitalization among regions.

As can be seen from Table 9, from the perspective of intra-group differences, the Gini coefficients in the three regions all show a downward trend of varying degrees. In particular, the intra-regional differences in the digitalization development of Europe show a particularly obvious downward trend, with a decrease of 23.7%, followed by Asia, where the intra-regional Gini coefficients in the digitalization development of Asia decrease by 9.5% from 2012 to 2021. The smallest decline, 8.1 per cent, was seen on both sides of the Atlantic. Meanwhile, the average intra-regional Gini coefficients of the three regions from 2012 to 2021 are ranked from lowest to highest, and it is found that the intra-regional difference is the smallest in Europe (0.0393), followed by both sides of the Atlantic (0.0754), and the largest intra-regional difference is in Asia (0.1138). Overall, the intra-regional Gini coefficient showed a fluctuating downward trend, indicating that the intra-regional imbalances in digitalization development in the seven countries have been improved. From a horizontal comparison of inter-regional differences,

**Table 9. Results of regional differences in digitalization development.**

| Year | Intra-group Gini coefficient | | | Inter-group Gini coefficient | | |
|---|---|---|---|---|---|---|
| | Asia | Both sides of the Atlantic | Europe | Asia & both sides of the Atlantic | Asia & Europe | Both sides of the Atlantic & Europe |
| 2012 | 0.12 | 0.169 | 0.03 | 0.174 | 0.105 | 0.183 |
| 2013 | 0.225 | 0.183 | 0.009 | 0.255 | 0.256 | 0.183 |
| 2014 | 0.153 | 0.099 | 0.138 | 0.214 | 0.199 | 0.143 |
| 2015 | 0.021 | 0.096 | 0.015 | 0.208 | 0.184 | 0.099 |
| 2016 | 0.181 | 0.002 | 0.056 | 0.191 | 0.191 | 0.056 |
| 2017 | 0.167 | 0.03 | 0.057 | 0.182 | 0.216 | 0.063 |
| 2018 | 0.06 | 0.064 | 0.024 | 0.104 | 0.179 | 0.094 |
| 2019 | 0.106 | 0.026 | 0.03 | 0.12 | 0.155 | 0.087 |
| 2020 | 0.061 | 0.012 | 0.032 | 0.198 | 0.159 | 0.042 |
| 2021 | 0.044 | 0.073 | 0.002 | 0.146 | 0.134 | 0.074 |

the average value of inter-regional differences in the global digital economy are ranked from small to large as both sides of the Atlantic and Europe (0.1024), Asia and Europe (0.1778), and Asia and both sides of the Atlantic (0.1792). It can be found that the inter-regional differences between Asia and both sides of the Atlantic are still large. In the future, it is still necessary to focus on improving the level of digitalization development in the Asian region in the future.

## 4.2 Difference analysis of digitalization development level

In order to further research the causes of the differences in the level of digitalization development, we first define the definition of digitalization level construction. The construction of digitalization level refers to the promotion of informatization and intelligent development through digital technology in various economic, social and political fields, to enhance the digital capacity and development level of organizations and society, including the construction of infrastructure, informatization management, and application systems, data management and analysis capacity, talent cultivation and innovation capacity, and other aspects. Furthermore, the construction of the digitalization level mainly relies on the integrated elements of technological innovation, data management and sharing, policy support, talent cultivation and development, as well as public participation and digital innovation awareness cultivation to promote it.

In the comprehensive development level of these factors, the reason why China and the above major developed countries have a big difference, an important influencing factor is that China and these countries are in different stages of economic development. The above-mentioned major developed countries have already entered into the digitalization technological economy paradigm earlier, but our country's industrialization started late, and have not fully entered the stage of digitalization technological economy paradigm in the industrial structure [34]. Therefore, in order to effectively promote China's digitalization development by upgrading the comprehensive development level of these elements, it is necessary to first vigorously promote the continuous innovation of the techno-economic paradigm, because different techno-economic paradigms will lead to different digitalization development paths. For example, the information technology revolution has become the dominant techno-economic paradigm in the past few decades, and the popularization and application of information technology have transformed people's lifestyles, business models, and social interactions, driving a new round of economic growth. In addition, different countries and regions have different techno-economic paradigms, and their impact on digitalization development varies. It can be seen that the techno-economic paradigm has an important impact on the level of digitalization development, and the evolution of the techno-economic paradigm promotes the improvement of the level of digital construction, which in turn supports the optimization of the new techno-economic paradigm to achieve high-quality and sustainable digitalization development. In order to further clarify this issue, this paper first combs, we also need to deeply sort out, review, and analyze the evolution of the techno-economic paradigm in major developed countries and China.

**4.2.1 The evolution of the techno-economic paradigm in the United States.**   The different stages of technological development in the United States and its techno-economic paradigm exhibit different characteristics, the details of which are shown in Table 10.

**4.2.2 The evolution of the British techno-economic paradigm.**   The first and second technological revolutions represented by the factory production of textiles and the era of steam power and railways, the United Kingdom has been at the forefront of the world and promoted the development of the world economy. In this process, the British government has gradually formed an innovation system composed of enterprises, universities, government, scientific

**Table 10. The evolution of the techno-economic paradigm in the United States.**

| Stages of technological development | Manifestations of the techno-economic paradigm at different stages |
|---|---|
| Origin stage | ①Respect for the freedom of enterprise and individual innovation and invention<br>②The government has successively promulgated the Patent Protection Act, the Moore Act, and the Adams Act.<br>③The initial establishment of a national system of scientific research institutions, comprising higher education institutions, enterprises, private research organizations, and federal laboratories.<br>④Science, technology, and innovation are predominantly in support of agriculture, with most of the scientific development and innovation research being undertaken by individuals and businesses, and with very limited government funding. |
| Exploration phase | ①Establishment of a scientific and technological innovation system focusing on military engineering and taking into account basic scientific research.<br>②Launch of the Manhattan Project, a government-led national R&D system with federal laboratories as the mainstay [35].<br>③We will increase national investment in scientific and technological research and development, and establish a national science foundation to support the rapid development of basic scientific research. |
| Mature stage | ①The implementation of the coordinated development strategy for military, civilian, and commercial innovation has formed a national scientific and technological innovation system that organically combines "government industry, academia, and research" [35].<br>②Adjust the ratio of federal R&D investment for defense and nondefense to 5:5 [35].<br>③Emphasize the education and cultivation of scientific and technological talents, and support innovation funding from private and small and medium-sized enterprises.<br>④Attaching great importance to the formulation and implementation of science, technology, and innovation development strategies, such as the biotechnology strategy, the "information superhighway" plan, and the national nanotechnology plan. |
| Developmental stage | ①The government has invested heavily in the National Institutes of Health and the Foundation Committee to reinforce scientific research activities; and has established the Commission on Science, Technology and Social Effectiveness to harmonize science and technology policies with social policies.<br>②Reform education, cultivate professional and technical talents, prioritize the development of clean energy, health care advanced automobile manufacturing, and other fields; Reform of the financial system.<br>③Encourage innovation in small and medium-sized enterprises and implement tax relief policies [36].<br>④Reform of government functions to create a service-oriented government. |

research institutions, and other non-profit organizations [37]. The techno-economic paradigm has evolved and developed from liberalization to government-guided, from innovation-enabling to innovation-service, and from the globalization of international cooperation to knowledge-based.

1. The techno-economic paradigm of liberalization
   Before the 1990s, the UK government took the approach of less meddling and non-interference except in defense science and technology.

2. Government-led techno-economic paradigm
   With the rapid development of the electronics industry, the impact of science and technology on the economy has become increasingly prominent. The United Kingdom has released its first technology white paper and elevated technological innovation to a national strategic level, emphasizing that the government should play a leading role [38]. The government started to increase investment in technological innovation, the implementation of

the patent box system, and actively guide innovation resources to the convergence of enterprises. At the same time, the British government put science and technology management, business and industry under the same department for management, and set up an independent United Kingdom Research and Innovation Agency (UKRI).

3. Innovation-enabling techno-economic paradigm
The government has set up science and technology-based think tanks for policy formulation, consultation and suggestions; The National Foundation for Science, Technology and the Arts regularly organizes various seminars and exchanges on a regular basis, carries out research on innovation methods and trends, and continuously finances innovation projects to foster the innovation capacity of enterprises. The Intellectual Property Office (IPO) is responsible for intellectual property policy formulation, system operation, maintenance, etc., to accelerate the transformation of scientific and technological achievements. In addition, the government has increased research and development relief and promoted research data sharing in an open environment.

4. Innovative service-oriented techno-economic paradigm
The Government has implemented the Knowledge Transfer Partnership Program (KTP) to support technology development, cooperation between public research institutions and enterprises, and to promote innovation through cooperation between industry and academia. The government has established the National Center for University-Industry Cooperation (NCUB) to build a digital platform for industry-academia cooperation; and has constructed science and technology innovation parks, university incubators, and university technology transfer offices to promote knowledge transfer and the transformation of scientific and technological achievements.

5. Globalized techno-economic paradigm of international cooperation
The country has proposed the construction of a globalized innovation network, and has signed strategic agreements on international scientific and technological cooperation with the United States, Canada, China, and other countries to establish close scientific research partnerships.

6. Knowledge-based techno-economic paradigm
The UK has formed a knowledge creation system composed of universities, non-governmental institutions public research institutions, etc. The seven major research councils, divided by different fields, are nongovernmental public institutions, managing a total of nearly 20 research institutes to promote the dissemination of science, the development of science and technology and its application in their respective fields, and to cultivate research-oriented personnel [39]. Public research institutions are owned by the government, and public funds mainly finance public scientific research tasks.

**4.2.3 The evolution of the French techno-economic paradigm.** With the development of science and technology, France has roughly gone through the traditional industrial period, the stage of state-directed economic development, the rise of scientific research and innovation, and the stages of independent innovation, and the evolution of the techno-economic paradigm has gone from traditional industry to state-directed economic development, then to the development of a knowledge-based economy and high technology, and ultimately to the era of the digital economy.

1. Techno-economic paradigm of the traditional industrial period
The traditional industrial period was mainly dominated by traditional industries, such as

textiles, metallurgy, and other sectors. Under this paradigm, factories were central to industrialization, mass production, and standardized production were characteristic, and the division of labor was clear. In addition, the traditional industrial period was characterized by an economic paradigm dominated by industrial capital and the centralization of large enterprises. In this period, although it possessed a certain degree of technological strength and innovation in some areas, overall there was still a gap compared with the industrially advanced countries.

2. Techno-economic paradigm of national guidance for economic development
After the Second World War, France adopted a state-directed economic development model. France divides national research institutions into two categories: "science and technology" and "industry and trade" [40]. The former is mainly responsible for basic research, applied basic research, and basic frontier cross-research, while the latter focuses on national defense, energy security, and other related fields to carry out application and development research and a small amount of basic research. At the same time, France established the "Office of the Secretary of State for Scientific Research and Technological Progress" to promote the orderly conduct of scientific research activities [41]. The government promotes the construction of heavy industry and infrastructure through the establishment and continuous improvement of the national science and technology program system, the system of scientific and technological decision-making and consulting system, evaluation, and scientific and technological achievements transformation system. During this period, France made important breakthroughs in the fields of aerospace, nuclear energy, and high-speed railroads.

3. Techno-economic paradigm in the era of knowledge economy
Since the 1980s, France has gradually shifted towards a knowledge-based economy and high-tech sectors. The government has actively promoted scientific and technological research and innovation, and increased its support for high-tech enterprises and scientific research institutions. At the same time, the government has attached great importance to education and research, invested in higher education and scientific research institutions, and cultivated and attracted a large number of outstanding talents. Subsequently, the country carried out reforms of the science and technology system, mainly in the following aspects: first, the Law on Scientific Research and Innovation was promulgated and implemented in 1999, establishing a two-way mobility mechanism for scientific researchers on staff at the legislative level; second, the National Agency for Scientific Research (ANRS) and the French Public Investment Bank (BFIP) were reorganized as two classified and independently functioning specialized funding agencies to strengthen the management and allocation of competitive scientific research funds; third, the S&T evaluation system has been reconstructed and innovated in terms of simplifying the evaluation procedure, reducing the number of evaluations and enriching the dimensions of evaluation indexes; fourth, a comprehensive S&T strategic plan has been enacted and implemented to plan the development of key areas such as biotechnology, energy and transportation, nanotechnology, etc.; fifth, in order to continuously improve the decision-making level of different government departments, the French government has successively set up the Strategic Steering Committee, the National Science Council and the Supreme Council for Science and Technology, which are three scientific and technological decision-making consulting organizations subordinate to and serving different government departments [42]. France also has world-renowned research and innovation centers, such as the French National Center for Scientific Research (CNRS) and the Ecole Technologie Superieure de Paris (ENS).

4. Techno-economic paradigm for the autonomous innovation phase
   The French government attaches great importance to technological research and development (R&D); and supports the innovative activities of enterprises and scientific research institutions by increasing investment in R&D funding. The French government promulgated the Law on the Orientation and Planning of Scientific Research and Technology Development, which clearly stipulated the proportion of public scientific research funding to the gross national product in the form of legislation, and determined the annual growth rate of scientific research investment in accordance with the development needs of the prevailing disciplinary fields, while emphasizing and continuously and steadily supporting the role of basic research [43]. The government encourages and fosters the development of high-tech industries in frontier areas such as digital economy, biotechnology, new energy, etc., and actively builds an innovation and entrepreneurship ecosystem. The government establishes incubators, accelerators, and entrepreneurship funds to provide support and resources for startups, as well as services such as financing, training and consulting, to help startups grow and expand rapidly. In addition, France encourages and strengthens cooperation and transnational science, technology, and innovation cooperation projects with international scientific and technological organizations, enterprises and research institutes, thereby promoting scientific and technological innovation and development in the country.

**4.2.4 The evolutionary process of Japan's techno-economic paradigm.** Japan has always taken "technological powerhouse" as the main line of economic development, and has developed from pure technology importation to improvement and localization of imported technology, to imitation and creation, and ultimately to independent creation.

1. Techno-economic paradigm at the stage of simple imitation
   At this stage, the introduction of technology mainly focuses on basic industries such as electric power, coal, iron and steel, with the help of imported technical equipment and technology to carry out the technological transformation of the original industries, the implementation of the strategy of "industrialization of the country", with the mass production of products as the main focus.

2. The techno-economic paradigm of the imitation stage of innovation
   Japan has strengthened the digestion and absorption of imported technologies, placing greater emphasis on the introduction of technologies at the experimental stage in order to be the first to put them into production and take the lead in capturing the international market. At this stage, Japan is implementing the strategy of "building a nation through trade", utilizing imported basic technologies for independent development and application to enhance its international competitiveness [44].

3. Techno-economic paradigm in the autonomous innovation phase
   At this stage, the development strategy of a "technology-based country" is implemented, and the science and technology policy focuses on independent innovation, continuously increasing the proportion of investment in basic research, establishing an open innovation system of government, industry, academia and research, and international and domestic collaboration, strengthening the government's investment in science and technology and education, cultivating technological breakthrough talents, and setting up the Intellectual Property Strategy Committee to protect intellectual property rights and to promote the development of innovation.

**4.2.5 The evolutionary process of Korea's techno-economic paradigm.** Korea has adopted a government-led technology development model, whereby the government encourages the introduction of foreign capital and technology, establishes government-industry-university-research collaborative innovation and development, and improves the level and efficiency of technology research and development. At the same time, it has increased tax incentives and funding for technological research and development, successfully realizing the leap from imitation innovation to independent innovation.

1. Techno-economic paradigm under authoritarianism
   In this period, the government introduced technology and innovation to catch up with the rest of the world. The government imported basic technologies from abroad in the areas of textiles, electric power, transportation and social infrastructure, formulated the Science and Technology Revitalization Plan, encouraged technology trade, attracted foreign direct investment (FDI), cooperated with foreigners in cooperation and development, and promoted imitation through importation in order to continuously cultivate and strengthen industrial advantages [45]. Taking into account the economic situation of the country, Korea implemented a heavy chemical policy from the 1960s to the 1970s, and the industry began to gradually transform from a labor-intensive industry and light industry to a heavy industry. In the 1980s, the government adjusted its science and technology development strategy from "building the country by industry" to "building the country by technology", emphasized science and technology education and training of innovative talents, strengthened the digestion, absorption, and re-innovation of imported technologies, promoted the cooperation between industries, universities and research institutes, increased the support for basic and applied research, encouraged and supported the development of research and development institutes by the enterprises, so as to continuously improve the research and development capability of local enterprises.

2. Techno-economic paradigm under democracy
   In this period of independent and continuous innovation, the government continuously deepened market reforms and gradually reduced its direct intervention in the private sector. In the 1990s, government departments were involved in research and development and encouraged enterprises to cooperate with foreign institutions and industry-academia-research institutes in R&D, while focusing on intellectual property rights protection for small and medium-sized enterprises and individuals. The government established independent basic science academies and set up institutions at home and abroad, actively encouraging the participation of foreign scholars. It set up national R&D centers, expanded universities, and continuously raised the level of treatment for national talents.

**4.2.6 The evolution of the German techno-economic paradigm.** The country is based on federalism and a market economy, and its technological development has gone through four stages: mechanization (Industry 1.0 stage), electrification (Industry 2.0 stage), informatization (Industry 3.0 stage), and intelligence (Industry 4.0 stage).

1. Techno-economic paradigm for Industry 1.0 stage
   The government emphasized higher education, made universities the centers of scientific research established seminars and laboratories, and focused on the development of engineering. The government imported British machinery and technology at a high price, which quickly gave rise to technologically oriented large enterprises, and increased the

development of basic sciences, which quickly made it one of the world's most advanced industrialized countries.

2. Techno-economic paradigm for Industry 2.0 stage
Severely affected by the war, Germany's scientific and technological funding was sharply reduced, and electrical automation developed rapidly through the establishment of laboratories within large enterprises to vigorously develop pillar industries such as iron steel and electricity. The government established public scientific research institutions, implemented centralized research, shared research results with the community, shared patent ownership with the government, enterprises conducted spontaneous research, and applied technology universities and vocational education schools and enterprises to train professionals.

3. Techno-economic paradigm for Industry 3.0 stage
The Government focused on the development of areas such as biology, electronic information, and nanotechnology, and provided financial support to enterprises, universities, and research institutions in conjunction with the private sector, focusing on the development of small and medium-sized innovative enterprises. For example, the government, financial institutions, and large enterprises jointly established a small and medium-sized enterprise technology development fund; and jointed research and development by small and medium-sized enterprises and universities. The Government encouraged universities to apply for patents and to participate in major national scientific research tasks, with enterprise innovation at the core. The Government, public scientific research institutions, universities, and research and funding institutions provided comprehensive support and services to enterprises.

4. Techno-economic paradigm for Industry 4.0 stage
The Government has continued to improve the innovation environment, strengthen the construction of technological infrastructure, promote cross-industry and cross-border dialogue, focus on the integration of industry and academia, the training of talents, and the internationalization of the scientific research system, which is mainly involved in new areas such as electric transportation, environmental protection industry, medicine, public security, information and communication technology.

**4.2.7 The main characteristics of China's techno-economic paradigm at various stages of development.** After more than 70 years of development since the founding of the People's Republic of China, China has risen from a "poor and useless" scientific and technological foundation to the world's second-largest economy; and has achieved leapfrog development of scientific and technological innovation capability. The specific content of the evolution of China's techno-economic paradigm in the context of planned economy, market economy, and globalization is shown in Table 11. Currently, digital technologies such as big data, cloud computing, blockchain, artificial intelligence, and other digital technologies provide strong power support for the integration, transformation and upgrading of traditional industries, and release new kinetic energy for economic growth.

## 5. Discussion

With regard to digitalization development, previous scholars have studied the differences in business digital maturity of enterprises from a micro level [48, 49]. The influence of enterprise digitization level on economic growth have also analyzed from the micro level [50]. Some scholars have measured the development level of digital economy in 122 countries around the

**Table 11. The evolution of China's techno-economic paradigm.**

| Evolution stage of the techno-economic paradigm | The main characteristics of the techno-economic paradigm | | |
|---|---|---|---|
| Techno-economic paradigm at the stage of technology introduction | Three large-scale technology introductions | In the 1950s, the Soviet Union assisted the construction of "156 projects" | ①Led by the government, 156 projects were mainly in the heavy industrial sector.<br>②Equipment and science and technology were introduced and the first technicians were trained with the help of the Soviet Union.<br>③Introduction of the Soviet model of science and technology system |
| Techno-economic paradigm at the stage of technology introduction | Three large-scale technology introductions | The "Four-Three Plan" during the "Cultural Revolution" | ①Introducing petrochemicals, synthetic materials, and complete sets of equipment, with a focus on solving the problems of food and clothing.<br>②Focus on hardware (machinery equipment) introduction |
| | | Large-scale technology introduction in the pre-reform and opening-up period | ①Concentrate on the introduction of knowledge-intensive, technology-intensive products and establishment of advanced factories, such as Baosteel, to fill the production gap.<br>②Predominantly single-piece or small-lot production |
| Techno-economic paradigm in the transition stage | ①Introduction of key equipment manufacturing technologies and techniques, mainly for industrialization and first modernization.<br>②Diversifying the channels of introduction, continuously increasing the investment in R&D, and introducing advanced management methods and systems, mainly by state-owned enterprises.<br>③Significant increase in the number of government-owned scientific research institutions and technical personnel.<br>④Promote the restructuring of scientific research institutions and the mobility of researchers to enterprises. | | |
| Techno-economic paradigm under the market economy system | ①Implementing the strategy of developing the country through science and education, a series of laws, regulations, and policy documents have been promulgated, such as the Law on Promoting the Transformation of Scientific and Technological Achievements.<br>②The enterprise has gradually formed an independent product development platform.<br>③Scientific research institutes were transformed into enterprises.<br>④Multinational corporations have entered technology and capital-intensive industries and have established research and development institutions in China. | | |
| Techno-economic paradigm in the phase of globalization | ①The government has deepened the reform of the market economic system and promoted the organic integration of higher education with science and technology, economy and culture through dual drive.<br>②Technology development presents integration, intelligence, miniaturization, and other characteristics.<br>③Industrial organizations are developing towards network and ecology.<br>④Constantly meeting individual needs | | |
| Intelligent and Autonomous Innovative Techno-Economic Paradigm | ①Data has become a basic resource and an important production factor, and intelligent digital infrastructure has become the leading direction of new infrastructure [46].<br>②Economies of scale and scope based on personalized user value definition, where the user is the value definer and participates in value creation.<br>③Multi-faceted integration and innovation empowered to become the main engine of economic development [47].<br>④The business model is based on the huge data of the organization, and emphasizes the development of technical capabilities and the continuous provision of cloud services to the outside world. | | |

world from the meso level [51]. On the basis of existing research results, this paper measures the digitalization development level of the world's major developed countries and China, conducts an in-depth analysis from the new perspective of techno-economic paradigm, and launches a study on digitalization development from the macro level. The academic significance and practical guidance significance of this study are mainly as follows: firstly, the in-depth analysis of the digitalization development level of the world's major developed countries and China from the perspective of techno-economic paradigm can provide new ideas and methods for understanding the characteristics and trends of digitalization development in different countries. Second, by conducting research on digitalization development from a macro level, it can reveal the influence and development trend of digitalization development on a global scale, which can help governments and enterprises formulate more effective digitalization development strategies. Finally, such research can also help promote international cooperation and exchanges, promote mutual learning and reference between different countries in the field of digital development, and jointly promote the process of global digitalization development.

The relationship between techno-economic paradigm and digitalization development: (1) The transformation and evolution of techno-economic paradigm is an important prerequisite for digitalization development. Firstly, emerging technologies such as artificial intelligence, big data, cloud computing and so on continue to emerge under the impetus of techno-economic paradigms, providing a broader development space for digitalization development. Secondly, different techno-economic paradigms will affect the direction and focus of digitalization development, such as the upgrading of traditional industries, the development of emerging industries, digital transformation, etc., which will have different development strategies and implementation paths under different paradigms. In addition, with the change of techno-economic paradigm, it has an impact on the industrial ecology and market pattern of digitalization development, the original industrial chain may be subverted and restructured, and the new industrial ecology and market pattern will be gradually formed in the process of digitalization development. (2) Digitalization development will in turn influence and shape the techno-economic paradigm. The wide application and innovation of digital technology will lead to the adjustment and change of industrial structure and promote the formation and development of new techno-economic paradigm. At the same time, digitalization development encourages enterprises and organizations to adopt digital technology in production, management, marketing and other aspects, thus changing and affecting the way the economy operates and develops. (3) Techno-economic paradigms and digitalization development interact with each other and together form the basis and support of the digital economy. Digitalization development needs to adapt to different techno-economic paradigms, while the evolution of techno-economic paradigms also stems from the needs and impetus of digitalization development.

## 6. Conclusions and suggestions

This paper constructs a comprehensive evaluation index system for digitalization development, and tests the constructed index system. The entropy method is used to estimate the digitalization development level of seven countries from 2012 to 2021. Then, the variation coefficient method, kernel density estimation, Dagum Gini coefficient and its decomposition method are used to analyze the regional differences in the digitalization development level between China and the world's major developed countries, and the main conclusions are drawn as follows:

1. China's digitalization development level is the lowest among the seven countries, and its development is relatively slow, and the imbalance of its digitalization development level is more obvious.

2. The United States ranked first in digitalization development among seven countries in 2021 and 2022, and its overall development level in the digitalization field was relatively balanced from 2012 to 2021.

3. China has the lowest scores in the areas of talents in the digital field, digital infrastructure construction, digitalization innovation capability and international competitiveness, and there is a big gap with the other six countries. At the same time, the country has significant imbalances in all four areas.

4. The overall differences in digitalization development among the seven countries show a narrowing trend, and the regional differences in digitalization development mainly come from inter-regional differences. Next, we should focus on the coordinated development of digitalization between regions.

5. The difference in digitalization development between the three countries of China, Japan and Korea and the United States and the United Kingdom is still large, and there is still a need to focus on improving the digitalization development level of China, Japan and Korea in the future.

Through international comparison, it can be seen that compared with the world's big and powerful countries in digitalization development, China's digitalization development still has gaps and shortcomings. Based on this, the following suggestions are provided to promote the digitalization development of our country:

1. Create a new channel for digitalization human resources training. With the arrival of the digital era, the first thing to change is the mode of thinking and cognition, shifting from simple linear thinking to systematic digital thinking with multiple elements, multiple subjects, and multiple levels of cross-fertilization. To cope with the digital challenges, only by strengthening basic research can we fundamentally solve key technological problems at source and achieve high-level technological self-reliance and self-improvement [52]. However, strengthening basic research is ultimately dependent on high-level human resource elements. The cultivation of high-level talents has the characteristics of a long cycle, high investment and slow effectiveness. This requires the establishment of a diversified and multi-level human resource element cultivation system from basic research, technology development and university-enterprise cooperation in industry, academia and research to the transformation of scientific and technological achievements, so as to realize the whole-chain human resource cultivation [53]. Firstly, linking society, universities, and enterprises to participate in the whole process, from discovering human resources to selecting and cultivating high-level human resources to technological research and achievement transformation. Through various methods such as part-time innovation, long-term deployment, short-term cooperation, and other ways and in combination with the actual needs, and taking the service effectiveness as an important reference for the evaluation of the title and promotion of the position; secondly, we need to increase the support and investment in basic scientific research business fees, which will be used for relevant aspects of national strategic needs, carry out research on cutting-edge scientific issues, and implement a lump sum system for the use of funds; thirdly, universities and enterprises should actively explore human resources cultivation models such as order-type, modern apprenticeship system and market demand-oriented, establish a database of university-enterprise cooperation and form a "1+1+N" diversified cooperation model. The fourth is to establish an inter-industry human resources cooperation platform and standardize the flow mechanism of human resources factors; the fifth is to innovate human resources evaluation methods, carry out classified

evaluation, and establish and improve reasonable and complementary institutional evaluation standards; the sixth is to increase the number of science and technology talents to carry out international scientific and technological exchanges and cooperation, and to support them to go to foreign high-level scientific research institutions for study and training, and carry out cooperative research [54]. The seventh is to improve the life and service guarantee of scientific and technological talents, such as taking appropriate ways to improve the salary, performance pay and income from the transformation of achievements of scientific and technological talents, exploring and establishing an academic leave system, and creating a favorable, relaxed and harmonious scientific research cultural environment.

2. Promote techno-economic paradigm innovation, build and develop a digital technology innovation system and digitalization techno-economic paradigm, and promote the upgrading of digitalization development level. As the potential of the old techno-economic paradigm has reached its boundary barriers, serious constraints on the development of digitalization have been formed. Therefore, we should promote the rapid development of digitalization through techno-economic paradigm innovation, and establish the predominance of digitalization techno-economic paradigm. In essence, this process is the formation and development of a new industrial revolution, which is manifested in the process of promoting the modernization of the industrial system by a new round of industrial revolution [55]. Digitalization development takes data as the key core element and is based on digital infrastructure and data resource system. Only by promoting the development of big data, artificial intelligence, blockchain and other industries through techno-economic paradigm innovation, promoting the deep integration of digital technology and the real economy, and empowering traditional industries, can we better improve the level of digitalization development. We will promote the entire industrial system and trade in services towards digitalization, networking, and intelligent development.

## Supporting information

**S1 Table.**
(DOCX)

## Author Contributions

**Conceptualization:** Zongyuan Huang, Miaomiao Qin.

**Data curation:** Miaomiao Qin.

**Formal analysis:** Zongyuan Huang, Miaomiao Qin.

**Investigation:** Zongyuan Huang, Miaomiao Qin.

**Methodology:** Miaomiao Qin.

**Supervision:** Zongyuan Huang.

**Visualization:** Miaomiao Qin.

**Writing – original draft:** Zongyuan Huang, Miaomiao Qin.

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
