## [Decision Letter · Decision Letter 0]

24 Oct 2023

PONE-D-23-30193International comparison of China's digitalization level and its enlightenmentPLOS ONE

Dear Dr. Huang,

Thank you for submitting your manuscript to PLOS ONE. After careful consideration, we feel that it has merit but does not fully meet PLOS ONE’s publication criteria as it currently stands. Therefore, we invite you to submit a revised version of the manuscript that addresses the points raised during the review process.

We look forward to receiving your revised manuscript.

Kind regards,

JOANNA ROSAK-SZYROCKA, Assistant Professor

Academic Editor

PLOS ONE

Journal Requirements:

Reviewers' comments:

Reviewer's Responses to Questions

**Comments to the Author**

1. Is the manuscript technically sound, and do the data support the conclusions?

Reviewer #1: No

Reviewer #2: No

2. Has the statistical analysis been performed appropriately and rigorously? 

Reviewer #1: Yes

Reviewer #2: Yes

3. Have the authors made all data underlying the findings in their manuscript fully available?

Reviewer #1: Yes

Reviewer #2: Yes

4. Is the manuscript presented in an intelligible fashion and written in standard English?

Reviewer #1: Yes

Reviewer #2: No

5. Review Comments to the Author

Reviewer #1: The topic you have chosen is of great importance in the current global landscape, as digitalization plays a significant role in shaping economies and societies. While the manuscript shows promise, there are several areas that require substantial improvement before it can be considered for publication. Below are my critical comments and suggestions for manuscript development.

It is essential to clearly state the specific aims of the study. What aspects of China's digitalization are being compared? A concise and focused research question will provide a clear direction for the readers.For example, why and how digitilization. Quantum computing holds the potential to revolutionize various fields, and one area where it can make a significant impact is in digitalizatio.Digitalization often involves optimizing processes, resources, and systems. Quantum computing excels at solving complex optimization problems. For example, it can optimize supply chain logistics, financial portfolios, energy distribution, and various other aspects of digitalized systems. 

-A robust methodology strengthens the credibility of the study's findings.The manuscript lacks an in-depth comparative analysis of China's digitalization level with other countries. Detailed comparisons, supported by statistical data and relevant literature, are essential to draw meaningful conclusions.

The conclusion section should summarize the key findings of the comparative analysis and their implications. Clearly state the contributions of your study to the existing literature. Discuss the practical implications for policymakers, businesses, and academics. Additionally, suggest avenues for future research based on the gaps identified during the comparative analysis. Ensure that each section flows logically, leading the reader from one point to the next seamlessly.Addressing these critical points will significantly enhance the quality and impact of your manuscript. I encourage you to revise your work thoroughly, taking these comments into consideration.

Reviewer #2: 1. The purpose of the article is specified in the abstract as" this paper evaluates the digital development level of China, the United States, the United Kingdom, France, and other major developed countries in the world based on the entropy method." Then, also at the and of introduction as: "In order to better achieve this strategic goal, we need to conduct a scientific and reasonable evaluation of China's digital development level, clarify the current situation of China's digital development, and conduct a scientific comparative analysis with major leading countries in the world, to find out the effective ways and countermeasures of China's digital development. - there is a kind of discrepancy and it should be corrected.

2. The title of the point 3 is too long "3. Comparison of digital development levels between China and major developed countries based on entropy method" . Moreover most of the part of it, is too much similar to the title of the article.

3. So long part of the article devoted to digitalization should be presented earlier in the introduction not in the methodological part

4. Conclusion is too long and this division into subparts are unnecessary. So, most part of conclusion should be included in the results part.

5. References list is limited.

6. PLOS authors have the option to publish the peer review history of their article (what does this mean?). If published, this will include your full peer review and any attached files.

Reviewer #1: No

Reviewer #2: No

---

## [Author Response · Author response to Decision Letter 0]

3 Dec 2023

Thank you very much for taking the time to review this manuscript. Those comments are all valuable and very helpful for revising and improving our paper, as well as the important guiding significance to our researches. We have studied comments carefully and have made correction which we hope meet with approval. Revised portion are marked in red in the paper. See the document "Responding to Reviewers".

---

## [Decision Letter · Decision Letter 1]

26 Dec 2023

PONE-D-23-30193R1International comparison of China's digitalization level and its enlightenmentPLOS ONE

Dear Dr. Huang,

Thank you for submitting your manuscript to PLOS ONE. After careful consideration, we feel that it has merit but does not fully meet PLOS ONE’s publication criteria as it currently stands. Therefore, we invite you to submit a revised version of the manuscript that addresses the points raised during the review process.

Based on the reviews received, please submit major revisions in accordance with the recommendations.

We look forward to receiving your revised manuscript.

Kind regards,

JOANNA ROSAK-SZYROCKA, Assistant Professor

Academic Editor

PLOS ONE

Reviewers' comments:

Reviewer's Responses to Questions

**Comments to the Author**

1. If the authors have adequately addressed your comments raised in a previous round of review and you feel that this manuscript is now acceptable for publication, you may indicate that here to bypass the “Comments to the Author” section, enter your conflict of interest statement in the “Confidential to Editor” section, and submit your "Accept" recommendation.

Reviewer #1: All comments have been addressed

Reviewer #3: (No Response)

2. Is the manuscript technically sound, and do the data support the conclusions?

Reviewer #1: Yes

Reviewer #3: Yes

3. Has the statistical analysis been performed appropriately and rigorously? 

Reviewer #1: Yes

Reviewer #3: Yes

4. Have the authors made all data underlying the findings in their manuscript fully available?

Reviewer #1: Yes

Reviewer #3: No

5. Is the manuscript presented in an intelligible fashion and written in standard English?

Reviewer #1: No

Reviewer #3: Yes

6. Review Comments to the Author

Reviewer #1: I find the topic of your research highly interesting, and I appreciate the effort you have put into presenting your work in a clear and concise manner. Your research is valuable to our readership, and I believe it will make a meaningful contribution to the field. The overall quality of your work, along with the thoroughness of your revisions, has led to its acceptance. I commend your dedication to improving the manuscript and effectively addressing the reviewers' comments.

Reviewer #3: The manuscript entitled "International comparison of China's digitalization level and its enlightenment" was reviewed.

The abstract is important to represent the content of the article and it need to be improved. For example, the first part of the abstract should be improved providing a better introduction of the topic and the reason that you conduct this work.

The introduction section: What is the novelty for this article compared with existing studies? Main objectives of the manuscript are not clearly mentioned. What is the scientific contribution of your paper to the science? Can you please indicate scientific implications of your paper?

After sentences "With the development of information technology, the important role of digitization in modern

economic development is becoming increasingly prominent. The so-called digitalization refers to a new

economic paradigm that utilizes digital technology to promote the high-quality and sustainable

development of the modern economy and society" add references: https://www.mdpi.com/2199-8531/8/2/70;
https://www.nature.com/articles/s41599-023-02415-1.

Literature review section: Please, provide an overview of other research that has done similar overview. Here are some examplese, please consider them for the literature review: https://www.mdpi.com/2199-8531/8/1/27;
https://journals.plos.org/plosone/article?id=10.1371/journal.pone.0254993;
https://journals.plos.org/plosone/article?id=10.1371/journal.pone.0253965;
https://www.mdpi.com/2071-1050/11/8/2204;
https://www.sciencedirect.com/science/article/abs/pii/S0160791X20302244.

I have never seen any research gap as a sub section (literature review section) , which is very important for proving the novelty of the paper.

Research methods:

Add data information. What year do they come from, what database, etc. How were the indicators selected?

The choice of your research methods should be elaborated. Why did you use the specific methods that you use? For which other research were they also used? Please, provide references of other research that use the similar methods. It should be clear to the readers why you have chosen the proposed methodology.

Add a Discussion section: It would be appropriate to specify in more detail how this research differs from the already published paper that deals with a similar topic. To increase the significance of the results, the discussion part should embrace the differences and similarities among your findings and those of other scholars.

Conslucions: The section Conclusion can be expanded to include the limitations of this work and an indication of future research directions. Further applications of this proposed method should be discussed in detail, it may also good to add some real life cases, which can express the benefit of proposed method, further, and it may catch more attention.

7. PLOS authors have the option to publish the peer review history of their article (what does this mean?). If published, this will include your full peer review and any attached files.

Reviewer #1: No

Reviewer #3: No

---

## [Author Response · Author response to Decision Letter 1]

3 Jan 2024

We thank the editors for arranging the review and the reviewers for their valuable comments. The authors have carefully responded to the questions as requested by the reviewers and academic editor, and have made careful revisions to the paper. Because of your suggestions, the revised manuscript is better and readers can get more valuable information. Thanks again to the academic editor and reviewers for your help.

---

## [Decision Letter · Decision Letter 2]

31 Jan 2024

PONE-D-23-30193R2International comparison of China's digitalization level and its enlightenmentPLOS ONE

Dear Dr. Huang,

Thank you for submitting your manuscript to PLOS ONE. After careful consideration, we feel that it has merit but does not fully meet PLOS ONE’s publication criteria as it currently stands. Therefore, we invite you to submit a revised version of the manuscript that addresses the points raised during the review process.

Please adapt the manuscript to the recommendations of the reviewers'. Apply the changes in yellow.

Please submit your revised manuscript by Mar 16 2024 11:59PM. If you will need more time than this to complete your revisions, please reply to this message or contact the journal office at plosone@plos.org. Please include the following items when submitting your revised manuscript:A rebuttal letter that responds to each point raised by the academic editor and reviewer(s). You should upload this letter as a separate file labeled 'Response to Reviewers'.A marked-up copy of your manuscript that highlights changes made to the original version. You should upload this as a separate file labeled 'Revised Manuscript with Track Changes'.An unmarked version of your revised paper without tracked changes. You should upload this as a separate file labeled 'Manuscript'.

We look forward to receiving your revised manuscript.

Kind regards,

JOANNA ROSAK-SZYROCKA, Assistant Professor

Academic Editor

PLOS ONE

Reviewers' comments:

Reviewer's Responses to Questions

**Comments to the Author**

1. If the authors have adequately addressed your comments raised in a previous round of review and you feel that this manuscript is now acceptable for publication, you may indicate that here to bypass the “Comments to the Author” section, enter your conflict of interest statement in the “Confidential to Editor” section, and submit your "Accept" recommendation.

Reviewer #3: (No Response)

Reviewer #4: (No Response)

Reviewer #5: All comments have been addressed

2. Is the manuscript technically sound, and do the data support the conclusions?

Reviewer #3: No

Reviewer #4: Partly

Reviewer #5: (No Response)

3. Has the statistical analysis been performed appropriately and rigorously? 

Reviewer #3: No

Reviewer #4: N/A

Reviewer #5: Yes

4. Have the authors made all data underlying the findings in their manuscript fully available?

Reviewer #3: No

Reviewer #4: No

Reviewer #5: Yes

5. Is the manuscript presented in an intelligible fashion and written in standard English?

Reviewer #3: Yes

Reviewer #4: Yes

Reviewer #5: Yes

6. Review Comments to the Author

Reviewer #3: Unfortunately, the authors did not respond to these comments. I do not recommend this manuscript for publication.

Reviewer #4: It would be appropriate to specify in more detail how this research differs from the already published paper that deals with a similar topic. To increase the significance of the results, the discussion part should embrace the differences and similarities among your findings and those of other scholars. You need to improve the practical and academic implications.

The discussion should compare China's performance with other countries, eg. European Union (https://www.mdpi.com/2199-8531/8/2/70, https://www.mdpi.com/2199-8531/8/1/27).

Reviewer #5: Ensure consistency in terminology and formatting throughout the paper. This includes the usage of headings, subheadings, and referencing.

7. PLOS authors have the option to publish the peer review history of their article (what does this mean?). If published, this will include your full peer review and any attached files.

Reviewer #3: No

Reviewer #4: No

Reviewer #5: No

---

## [Decision Letter · Decision Letter 3]

18 Apr 2024

International comparison of China's digitalization level and its enlightenment

PONE-D-23-30193R3

Dear Dr. Zongyuan Huang,

my congratulations! your manuscript has been judged scientifically suitable for publication and will be formally accepted for publication once it meets all outstanding technical requirements.

Kind regards,

JOANNA ROSAK-SZYROCKA, Assistant Professor

Academic Editor

PLOS ONE

---

## [Editor Report · Acceptance letter]

3 May 2024

PONE-D-23-30193R3 

PLOS ONE

Dear Dr. Huang, 

I'm pleased to inform you that your manuscript has been deemed suitable for publication in PLOS ONE. Congratulations! Your manuscript is now being handed over to our production team.

Kind regards, 

on behalf of

Dr. JOANNA ROSAK-SZYROCKA 

Academic Editor

PLOS ONE